# Efficient solar hydrogen generation in microgravity environment

Katharina Brinkert [1,2], Matthias H. Richter[1,3], Ömer Akay[4], Janine Liedtke[2], Michael Giersig[4,5], Katherine T. Fountaine[6,7] & Hans-Joachim Lewerenz[8]

Long-term space missions require extra-terrestrial production of storable, renewable energy. Hydrogen is ascribed a crucial role for transportation, electrical power and oxygen generation. We demonstrate in a series of drop tower experiments that efficient direct hydrogen production can be realized photoelectrochemically in microgravity environment, providing an alternative route to existing life support technologies for space travel. The photoelectrochemical cell consists of an integrated catalyst-functionalized semiconductor system that generates hydrogen with current densities >15 mA/cm$^2$ in the absence of buoyancy. Conditions are described adverting the resulting formation of ion transport blocking froth layers on the photoelectrodes. The current limiting factors were overcome by controlling the micro- and nanotopography of the Rh electrocatalyst using shadow nanosphere lithography. The behaviour of the applied system in terrestrial and microgravity environment is simulated using a kinetic transport model. Differences observed for varied catalyst topography are elucidated, enabling future photoelectrode designs for use in reduced gravity environments.

[1] Division of Chemistry and Chemical Engineering, California Institute of Technology, 1200 E California Blvd., Pasadena, CA 91125, USA. [2] Advanced Concepts Team, European Space Agency, ESTEC, Keplerlaan 1, Noordwijk 2200 AG, The Netherlands. [3] Brandenburg University of Technology Cottbus, Applied Physics and Sensors, K.-Wachsmann-Allee 17, 03046 Cottbus, Germany. [4] Department of Physics, Freie Universität Berlin, Arnimallee 14, 14195 Berlin, Germany. [5] International Academy of Optoelectronics at Zhaoqing, South China Normal University, 526238 Guangdong, China. [6] Resnick Sustainability Institute, California Institute of Technology, Pasadena, CA 91125, USA. [7] NG Next, Northrop Grumman Corporation, One Space Park, Redondo Beach, CA 90278, USA. [8] Division of Engineering and Applied Science and Joint Center for Artificial Photosynthesis, California Institute of Technology, 1200 E. California Blvd., Pasadena, CA 91125, USA. Correspondence and requests for materials should be addressed to K.B. (email: brinkert@caltech.edu) or to H.-J.L. (email: lewerenz@caltech.edu)

O ur atmosphere on earth is sustained by the photo-dissociation of water, a fundamental process used by nature in oxygenic photosynthesis to convert solar energy into storable, chemical energy[1]. It relies on the so-called Z-scheme, where photons of different energy are absorbed in a staggered energy-level system. Presently, artificial photosynthesis systems are being intensively developed[2–6] based on the analogue of the Z-scheme, e.g. in tandem solar cells[5]. Here, the light-driven oxidation of water to oxygen at the (photo) anode is accompanied by the production of so-called solar fuels at the (photo)cathode: hydrogen, a storable fuel which can be used as a feedstock for fuel cells generating power for transportation or carbon dioxide reduction products, holding the promise for converting emissions back to fuels utilizing renewable energy. Inorganic systems have yielded the hitherto highest efficiencies and stabilities when combining the tandem absorbers with high-activity electrocatalysts[7,8].

The efficient conversion of abundant sunlight to oxygen and storable fuels is also a key step in realizing long-term space missions and *cis*-lunar research platforms such as the Deep Space Gateway. Since the early 1960s, water electrolysis cells comprising of photovoltaic p–n junction solar cells and a separate water electrolyser system have been employed as a part of a spacecraft environmental control system for the production of oxygen from carbon dioxide[9] and are still in use on the International Space Station (ISS). The involved travel distances on future deep space missions restrict volume and mass of consumables required for a voyage of months or years, with a resupply of water and fuel from Earth becoming impossible. These long-duration trips into space demand regenerative, reliable and light life support hardware which repeatedly generates and recycles essential, life sustaining elements required by human travellers. An efficient and stable monolithic surface modified tandem device structure, capable of oxidizing water and simultaneously producing hydrogen and/or reducing $CO_2$ presents a compact and lighter alternative to the currently employed photoelectrolysis system. Moreover, analyses of terrestrial systems show that the fully integrated devices compare favourably with separate PV-electrolyser units regarding installation and fuel production costs[10,11]. Despite these advantages, direct photoelectrochemical water splitting for hydrogen and oxygen production[12] in space relevant conditions such as reduced gravitation has not been explored yet, although microgravity environment has already been employed for the electrochemical synthesis of advanced nanomaterials for energy conversion[13].

Herein, we describe the development of an efficiently operating semiconductor–electrocatalyst half-cell in microgravity environment. Experimental conditions are described which are required for investigating photoelectrochemical hydrogen production in reduced gravitational environments, realized at the Bremen Drop Tower. We show that due to missing buoyancy, microgravity has a significant impact on the mass transfer rate of protons to and hydrogen from the photocathode surface due to the formation of so-called gas bubble froth layers. These froth layers are known to drastically reduce ion and gas transport at electrodes in microgravity and increase the ohmic resistance in proximity to the electrode surface[14,15]. Using shadow nanosphere lithography (SNL)[16,17], we adjust the shape of the employed electrocatalyst on the photocathode and demonstrate continuous bubble release even at high current densities in the absence of buoyancy. The transfer of our concepts regarding catalyst micro- and nanotopography supported by theoretical analyses can contribute to the design of efficient life revitalization systems and energy generation in future space missions. Furthermore, they can be implemented in fully integrated devices for unassisted water splitting currently being realized for terrestrial applications.

## Results

**Drop tower experimental arrangement.** The investigation of light-induced hydrogen production on photocathodes was carried out at the Bremen Drop Tower, Center of Applied Space Technology and Microgravity (ZARM)[18], in a minimum $g$ level of $10^{-6}g$ with a free fall duration of 9.3 s (Fig. 1a). The complete photoelectrochemical potentiostatic experiment, comprising light sources, electrochemical cells, potentiostats, including analysis and recording devices was installed in a drop capsule (Fig. 1b) and submitted to microgravity in a free-flight drop tower experiment. A hydraulically controlled pneumatic piston-cylinder system launched the capsule upwards from the bottom of the tower. The capsule was accelerated in 0.25 s to a speed of 168 km/h closely to the top of the drop tube and then fell down into a deceleration chamber. The equipment of the 1.34 m tall drop capsule consisted of the photoelectrochemical setup with a two-compartment photoelectrochemical cell (Fig. 1c) which allowed the simultaneous investigation of two photoelectrodes during free fall. Two digital cameras recorded the gas bubble evolution behaviour on the photoelectrode surface for each cell compartment from the front and from the side. In order to avoid sample contact with the electrolyte prior to reaching microgravity conditions, a pneumatic lifting ramp was installed at the backside of the cell which allowed immersing and emersing the photoelectrodes upon command.

A programmed, automated drop sequence (see upper part in Fig. 1 and Supplementary Table 1 for more detailed information), ensured the precise synchronization of potentiostats, light sources, cameras and the pneumatic lifting ramp during the experiment.

Two different types of photoelectrodes were investigated in microgravity environment: in the first set of electrodes, p-type indium phosphide was employed as the photocathode material onto which rhodium particles were photoelectrochemically deposited under stroboscopic illumination (further on referred to as 'thin film electrodes')[19,20]. In the second set, SNL was applied to obtain nanocrystalline Rh particles of three-dimensional arrangement[16] on the p-InP photocathode. This technique using the self-assembly of hexagonal closed-packed monolayer of latex spheres was applied on the p-InP electrode to create masks for the electrodeposition of rhodium. After the latex spheres were removed, hexagonal unit cell patterns of the rhodium electrocatalyst were obtained (further on referred to as 'nanostructured electrodes').

**Photoelectrochemical behaviour.** Figure 2a shows the *J–V* measurements of the two different photocathodes types in terrestrial (1*g*) and microgravity conditions ($10^{-6}g$) in 1 M $HClO_4$ and in the presence of 1% isopropanol, which was added to the electrolyte to lower the surface tension and to favour gas bubble release[21]. Although both photoelectrodes exhibit similar *J–V* behaviour in terrestrial conditions, conspicuous differences are observed in the photocurrent–voltage characteristics in microgravity. The short circuit current of the thin-film sample was reduced by almost 70% during free fall, whereas the open circuit voltage decreased by 25%. Differences in the $V_{OC}$ of the nanostructured and thin-film sample in terrestrial conditions could have been attributed to performance differences of the photoelectrodes as shown in Supplementary Table 2. In contrary to the performance loss of the thin film in microgravity, the terrestrial *J–V* characteristics and also the cell efficiency remained the same when the nanostructured p-InP–Rh electrodes where exposed to microgravity (Fig. 2a). Similar observations were made in chronoamperometric measurements of the thin-film and nanostructured sample (Supplementary Figure 1) in microgravity

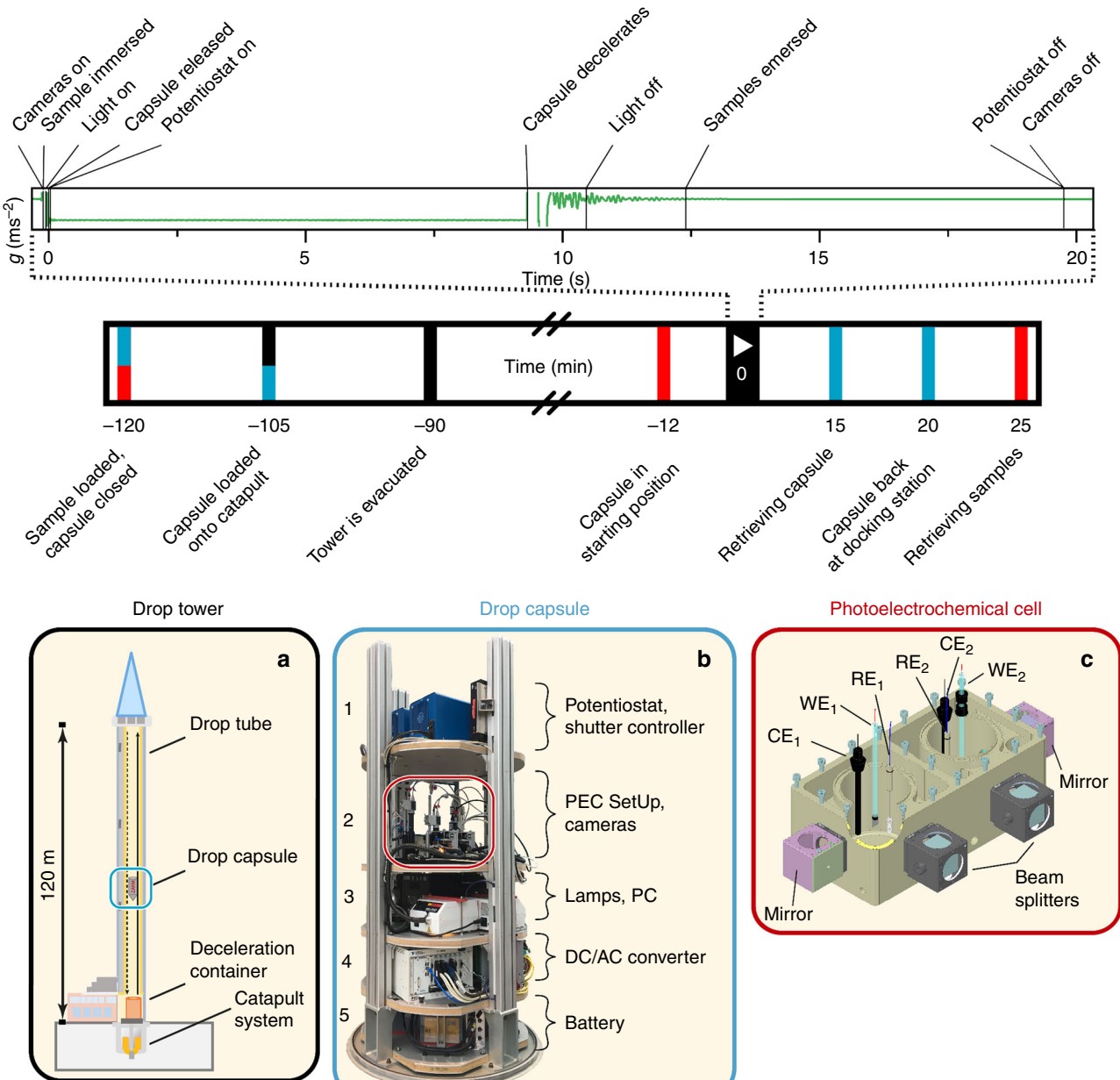

**Fig. 1** Scheme of the experimental set-up and time line of the photoelectrochemical experiments in microgravity conditions. The inset images show the drop tower (**a**), the drop capsule (**b**) and the photoelectrochemical cell (**c**). The capsule contained two potentiostats and two shutter control boxes (platform 1), the photoelectrochemical setup (platform 2) including four digital cameras, two W-I light sources and a Matrox 4Sight GPm computer (platform 3), a DC/AC converter (platform 4) and a battery for power supply during free fall (platform 5). The photoelectrochemical setup of platform 2 contained four digital cameras which allowed recording of gas bubble formation on the photoelectrode from the front through beam splitters and from the side through mirrors of the photoelectrochemical cell. Illumination of the photoelectrodes occurred through the beam splitters in front of the cell. A pneumatic lifting ramp ensured the immersion of the photoelectrodes in the electrolyte immediately before activation of the catapult system. WE is the working electrode, RE is the reference electrode and CE is the counter electrode. The subscript numbers indicate the respective cell compartment. The time line represents the programmed drop sequence. The inset shows the gravitational field according to the time line of the experiment. The colour code matches the events in the time line with the involvement of drop tower, drop capsule and/or photoelectrochemical cell

environment: the current density of the thin-film photoelectrode remained at a constant value of 5 mA/cm², whereas the current density of the nanostructured sample showed a nearly stable value of 16 mA/cm². This finding is also reflected in the gas bubble evolution behaviour on the two electrode surfaces (Fig. 2b): video recordings during the experiments reveal that although the electrodes do not show noticeable differences in their hydrogen evolution behaviour in terrestrial conditions, their microgravity behaviour differs significantly: here, the evolved hydrogen gas is

not released from the thin-film electrode surface and bubbles coalesce in proximity to the electrode, whereas the gas bubble release is enhanced on the nanostructured photoelectrodes. Similar observations have been made in various studies of water electrolysis in microgravity environments[9,15,22,23], where the absence of buoyancy and the suppression of natural convection caused the coalescence of gas bubbles on the electrode surface and the formation of froth layers. The resulting mass transfer limitations and the increased ohmic drop in the gas bubble

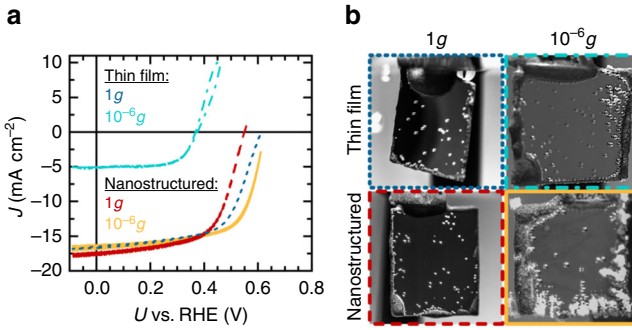

**Fig. 2** Results of the photoelectrochemical experiments in microgravity environment. **a** *J–V* measurements of thin-film and nanostructured p-InP–Rh photoelectrodes in terrestrial (1*g*) and microgravity environments (10⁻⁶*g*) at 70 mW/cm² illumination with a W-I lamp in 1 M HClO₄ with the addition of 1% (v/v) isopropanol. Differences in the $V_{OC}$ of the nanostructured and thin-film sample in terrestrial conditions are subject to performance differences of the photoelectrodes as shown in Supplementary Table 2. **b** Images from video recordings of the thin-film and nanostructured photoelectrodes after 9.3 s in terrestrial and microgravity conditions. In microgravity environment, hydrogen gas bubbles form a froth layer on the thin-film electrodes whereas the bubble adhesion to the electrode surface is decreased in the presence of the nanostructured Rh layer

dispersion zone in proximity to the electrode led to a substantial decrease in current density.

**Influence of surface nanotopography**. In order to elucidate the role of surface morphology, structure and composition of the thin-film and nanostructured photoelectrodes for their performance in microgravity environment, surfaces were characterized by structural, surface morphological and compositional analyses using atomic force microscopy (AFM), scanning electron microscopy (SEM), high-resolution transmission electron microscopy (HRTEM), energy-dispersive X-ray analyses and X-ray photoelectron spectroscopy (XPS). The photoelectrodeposition of rhodium on p-InP resulted in a homogenous layer of a rhodium grain conglomerate, whereas the application of SNL on p-InP prior to Rh electrodeposition resulted in a nanosized, two-dimensional periodic Rh structure (Fig. 3a–c).

SEM studies confirmed the homogenous array of holes in the metallic Rh film, in which rhodium exhibited a nanocrystalline cubic structure (Supplementary Figs. 2, 3). For both electrodes, XPS spectra (Supplementary Fig. 4) provide evidence for an InO$_x$/PO$_x$ oxide layer formation on the InP which is more distinct in the case of the nanostructured electrode, apparent in the larger InP signal at 128.4 eV. This is not surprising, given the fact that the PS bead prepared structure leaves open areas of InP which is accessible by the electrolyte. Despite the fact that Rh was deposited through the polystyrene sphere mask, the Rh $3d_{3/2}$ and $3d_{5/2}$ signal intensities are almost identical, suggesting a similar overall coverage of Rh on both electrodes with distinctive differences in the local coverage due to the different surface topographies.

## Discussion

In order to further understand the processes involved in the current–voltage reduction of the thin-film samples in microgravity environments observed here, a terrestrial experiment was designed, demonstrating the involvement of mass transfer limitations in the current density drop (Supplementary Fig. 5a): the photoelectrochemical cell with the photoelectrode was placed upside down and illumination occurred from the bottom of the cell via an optical mirror. This set-up allowed trapping the

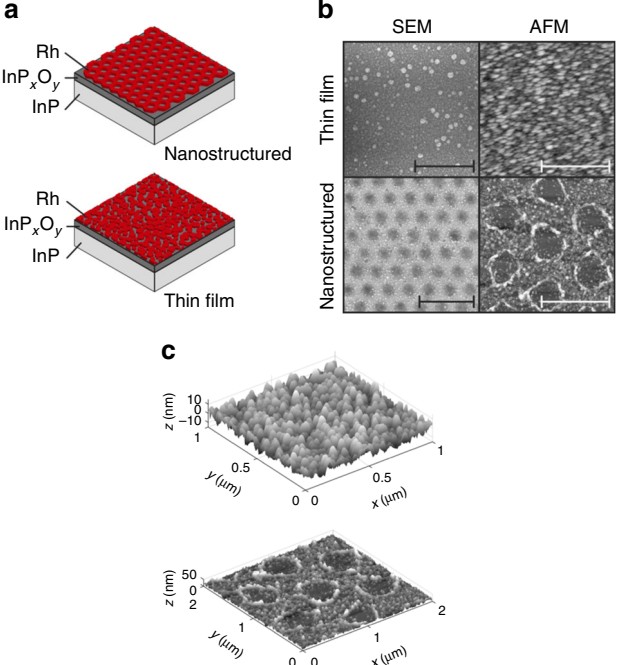

**Fig. 3** Structural investigations of the thin-film and nanostructured photoelectrodes. **a** Scheme of the two photoelectrode sets which were investigated in microgravity environment. In both cases, p-InP was employed as the light-absorbing semiconductor coated with a rhodium electrocatalyst layer. In the first set of electrodes, rhodium was photoelectrochemically deposited onto the planar p-InP surface. In the second set, Rh was deposited onto the p-InP surface through a mask of polystyrene particles, resulting in a hexagonal unit cell pattern of the rhodium after removal of the latex spheres. **b** SEM and tapping mode AFM images of the planar and nanostructured catalytic layer of rhodium on p-InP (also compare Supplementary Fig. 4c). The scale bars indicate the resolution of 500 nm of the thin-film electrode SEM and AFM images and 2 μm and 1 μm for the nanostructured electrode SEM and AFM images, respectively. **c** Three-dimensional surface structures of the thin-film and nanostructured photoelectrode obtained by AFM used for the calculation of catalyst surface area in the simulations

produced gas bubbles on the electrode surface while simultaneously recording the *J–V* characteristics. After initiating the hydrogen evolution reaction, the photocurrent density dropped instantaneously, resulting in an overall decrease of about 25% after 25 min reaction time. The open circuit voltage was also reduced by 50 mV (Supplementary Fig. 5b). The initial *J–V* behaviour could be recovered again when the surface tension of the electrolyte was decreased by addition of 1% (v/v) isopropanol, causing an enhanced gas bubble detachment from the electrode surface.

To elucidate the role of mass transfer for the performance of the thin-film electrode in microgravity environments further, the *J–V* characteristics of the investigated devices were theoretically modelled in terrestrial and reduced gravity environments. A semi-analytic formalism for PEC devices with nanostructured catalysts was used that builds on our model developed in previous publications[7,8,24] to include mass transport limitations.

The full current–voltage characteristics of the device when the rate of reaction is determined solely by reaction kinetics and without any mass transport considerations is captured via Eq. (1). It is an analytic equation for the current–voltage behaviour of a nanostructured coupled electrocatalyst–semiconductor device, in which *k* is the Boltzmann constant, *T* (K) is the temperature,

which is assumed to be 300 K, $q$ is the elementary charge, $j_L$ is the light limited current, $j_0$ is the dark current, $R$ is the universal gas constant, $n_e$ is the number of electrons associated with reaction which is 2 in this case, $F$ is Faraday's constant, $j_{0,cat}$ is the catalyst exchange current density and $f_{SA}$ is the catalyst surface area factor relative to the planar device area.

$$V_{PEC}(j) = \frac{kT}{q}\ln\left(\frac{j_L - j}{j_0} + 1\right) - \frac{2RT}{n_e F}\sinh^{-1}\left(\frac{j}{2j_{0,cat}f_{SA}}\right) \quad (1)$$

This equation is essentially the difference between the photovoltage of the diode, derived from the ideal photodiode equation, and the overpotential of the catalyst, derived from the Butler–Volmer equation[24].

When mass transport plays a role in the reaction rate, the current of the reaction can be derived from the Koutecky–Levich equation, Eq. (2), in which $j_{rr}$ is the overall current of the reaction, $j_{BV}$ is the kinetic current and $j_{mtl}$ is the mass transport current.

$$\frac{1}{j_{rr}} = \frac{1}{j_{BV}} + \frac{1}{j_{mtl}} \quad (2)$$

Essentially, this equation represents the kinetic and mass transport currents as two parallel current pathways. The kinetic current is described by Butler–Volmer kinetics, as in Eq. (1) for the non-mass transport limited case. The mass transport current is described by Eq. (3), in which $j_{mtl,a/c}$ are the anodic and cathodic limiting mass transport current densities, respectively, $e$ is the elementary charge and $V_{mt}$ is the overpotential due to mass transport.

$$j_{mtl}(V_{mt}) = \left(1 - \exp\left(-\frac{n_e e V_{mt}}{kT}\right)\right) \cdot \left(\frac{1}{j_{mtl,a}} - \frac{1}{j_{mtl,c}} \cdot \exp\left(-\frac{n_e e V_{mt}}{kT}\right)\right)^{-1} \quad (3)$$

This equation is derived from Fick's 1st law for diffusion across the Nernst boundary layer (Supplementary Fig. 6a) and the Butler–Volmer equation. A full derivation of this equation can be found in Supplementary Information (Supplementary Note 1). The current–voltage curve can then be found numerically by first calculating the $\{j_{PV}, V_{PV}\}$ and $\{j_{rr}, V_{mt}\}$ pairs from the ideal photodiode equation (first term in Eq. (1)) and the Koutecky–Levich equation, Eq. (2), respectively, using equal current values for both sets of pairs; and secondly, by subtracting the mass transport overpotential, $V_{mt}$, from the diode photovoltage, $V_{PV}$.

This process is the numerical equivalent to Eq. (1), in which the second term is replaced with the Koutecky–Levich equation, Eq. (3), (Supplementary Fig. 6b). Figure 4 summarizes the simulated results. Whereas nanostructured and thin-film sample shows a nearly equivalent $J$–$V$ behaviour in terrestrial environments, the assumed mass transfer limitations in microgravity environment affect the performance of the thin-film electrode significantly, providing strong evidence that this is the main effect leading to the decrease in photocurrent during free fall. The $V_{OC}$ decrease of the thin-film sample is furthermore a result of the light-induced excess electron accumulation at the photoelectrode surface which originates from limited mass transfer causing increased recombination at the electrode surface due to charge transfer inhibition[25]. It is reflected in the simulation by a lower dark current value, $j_0$ (see Methods part). Generally, it is to be taken into account that for the simulations, the same catalytic activity for Rh ($j_{0,cat}$) is assumed in terrestrial and microgravity environments. Due to the formation of gas bubble froth layers, some catalytic Rh sites might not be active in microgravity conditions, leading to a lower value for $j_{0,cat}$ and furthermore, a slower initiation of catalysis close to the $V_{OC}$. Furthermore, non-

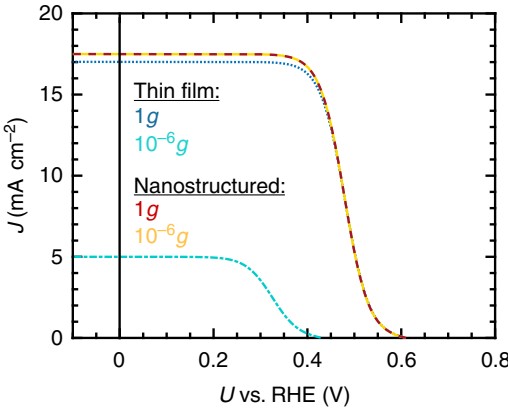

**Fig. 4** Simulations of the $J$–$V$ characteristics of the thin-film and nanostructured p-InP–Rh photoelectrodes in terrestrial and microgravity environments. Illumination was assumed to occur at 70 mW/cm² through a W-I lamp. The electrolyte composition was 1 M HClO₄ with the addition of 1% (v/v) isopropanol. The microgravity environment was artificially created by assuming that the $J$–$V$ characteristics of the thin-film photoelectrode are mass transfer limited (see text for details). The dashed red line corresponds to the nanostructured sample in 1$g$, the subjacent yellow line corresponds to the sample in $10^{-6}g$. The blue dotted line shows the behaviour of the thin-film sample under terrestrial conditions whereas the cyan dotted-dashed line corresponds to the $J$–$V$ characteristics in $10^{-6}g$

idealities in the photodiode equation which reduce the fill factor, such as series resistance (resistance across the InPO$_x$ layer) and shunt resistance (incomplete junction) accounting for e.g. interface resistance, are not considered in the simulations. Although previously considered[8], they were neglected in the description of the $J$–$V$ behaviour of this system since a variety of variables influence these experimental parameters which then only operate as additional fitting parameters.

The results show that under microgravity conditions, the electrode surface morphology plays a crucial role for the photoelectrochemical performance. The catalyst micro- and nanotopography have decisive influence on the life cycle of bubbles on the surface. The growth and accumulation of bubbles on the thin-film electrode leads to a froth layer also observed in dark electrolysis experiments[9,15,22,23] that seriously inhibits the hydrogen evolution reaction. Figure 5a sketches the effect of lateral accumulation of gas bubbles that form a gaseous interphase which increasingly suppresses hydronium ion transport to the surface. However, with specific nanotopographies one can overcome microconvectional limitations[26]: the three-dimensional catalyst structure, depicted schematically in Fig. 5b, generates hot spots due to increased local electrical fields at the tips of the structure that has been formed by SNL. Bubble generation occurs preferably at the tips of the catalyst structure. Figure 5b illustrates the effect: gas bubbles nucleate and grow at the tips of the Rh deposits that have been formed at the circumference of the open InP circles. The removal of the grown bubbles results from weakened adhesion to the surface due to the small contact area in conjunction with microconvection. Concentration gradients along the surface facilitate H₂ transfer to the bubbles upon formation. In addition to the decreased probability of forming bubble agglomerates on the electrode surface, the bubble size is further determined by the stability of the formed gas bubble on the Rh tip. These morphologic advantages lead to an increased $J$–$V$ performance in microgravity and suggest a first design principle for photoelectrodes employed in this environment for light-assisted fuel production.

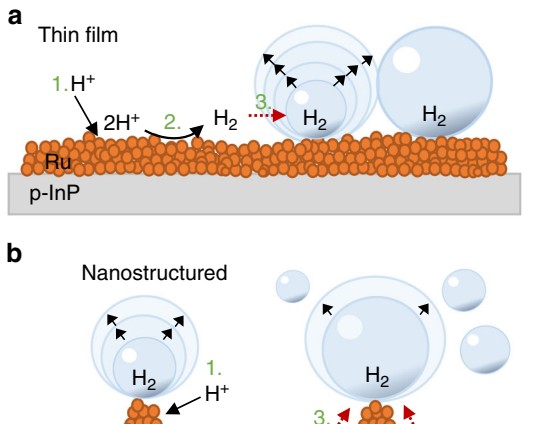

**Fig. 5** Cross sectional illustration of a gas bubble evolution model on the thin-film and nanostructured photoelectrode. Whereas $H_2$ is formed at discretionary nucleation spots on the thin-film electrode surface (**a**) resulting in gas bubble coalescence and the formation of a bubble froth layer, the nanostructured Rh surface favours the formation of $H_2$ gas bubbles at the induced Rh tips, catalytic hot spots (**b**). Here, concentration gradients along the surface facilitate $H_2$ transfer to the bubbles upon formation. The distance between the hot spots prevents the coalescence of the formed gas bubbles

We report efficient light-generated production of hydrogen on InP–Rh photoelectrodes in microgravity environment by performing photoelectrochemistry during drop tower flights. The reduction of the photocurrent due to the absence of buoyancy and therefore inhibited bubble removal is observed on photocathodes with unstructured, rather planar electrocatalysts (thin-film electrodes). Mathematical modelling shows—in a straightforward extension of the Butler–Volmer equation combined with a description of coupled diode–electrocatalyst systems—that mass transport limitations in the electrolyte result in suppression of the respective photocurrents from the thin-film samples. The preparation of a judiciously chosen nanostructured catalyst surface topography produces highly active Rh 'hot spots' preventing bubble coalescence and also favouring the detachment of produced hydrogen gas bubbles from the electrode surface. The developed model reproduces both, the photocurrent–voltage behaviour of nanotopographic and planar-type thin-film samples.

SNL has been used for the design of specific, three-dimensional nanostructures establishing the method as an attractive general tool for the design of high-activity photodiode–electrocatalyst systems. Our demonstrated efficiently operating half-cell producing hydrogen in microgravity environment opens up an alternative pathway for the improvement and extension of life support systems for long-duration space travel and *cis*-lunar research platforms while promoting investigations of currently developed systems for terrestrial solar fuel production for application in space. Investigations of phenomena such as gas bubble formation and evolution in reduced gravity environments can furthermore also lead to an enhanced understanding of processes at the electrode–electrolyte interface in photoelectrochemical devices, complementing ongoing terrestrial studies.

## Methods

**Preparation of the p-InP photoelectrodes**. Single crystal p-InP wafers with the orientation (111A) were obtained from AXT Inc. (Geo Semiconductor Ltd.)

Switzerland) with a Zn doping concentration of $5 \times 10^{17}\ cm^{-3}$. The preparation of an ohmic back contact involved the evaporation of 4 nm Au, 80 nm Zn and 150 nm Au on the backside of the wafer which was then heated to 400 °C for 60 s. The 0.5 $cm^2$ polished indium face of (111A) p-InP was furthermore etched for 30 s in bromine (0.05% (w/v))/methanol solution, rinsed with ethanol and ultrapure water and dried under nitrogen flux. All solutions were made from ultrapure water and analytical grade chemicals with an organic impurity level below 50 ppb. Subsequent cyclic voltammetric and chronoamperometric measurements were performed in a standard three-electrode potentiostatic arrangement whereas a carbon electrode was used as counter electrode and an Ag/AgCl (3 M) was employed as reference electrode. All potentials are converted to those vs. reversible hydrogen electrode (RHE). Moreover, the p-InP surface was photoelectrochemically conditioned in 0.5 M HCl, realized by potentiodynamic cycling under illumination (100 mW/cm$^2$) between −0.44 V and +0.31 V at a scan rate of 50 mV/s while purging with nitrogen of 5.0 purity. Illumination occurred with a white-light tungsten halogen lamp (Edmund Optics) through a quartz window of the borosilicate glass cell. The light intensity was adjusted with a calibrated silicon reference photodiode.

A thin Rh layer was photoelectrochemically deposited from a solution of 5 mM RhCl$_3$, 0.5 M NaCl and 0.5 vol% 2-propanol for 5 s at a constant potential of $V_{dep}$ = + 0.01 V and a light intensity of 100 mW/cm$^2$ using the same settings as for the photoelectrochemical conditioning procedure. The electrodeposition resulted in the formation of a nanocrystalline thin film or a nanostructured surface morphology if the rhodium was deposited through a polystyrene mask applying SNL (see below).

To compare the current–voltage characteristics and solar-to-hydrogen conversion efficiency of the photocathodes under terrestrial and microgravity conditions, sample electrodes were also tested in 1 M HClO$_4$ electrolyte solution upon illumination with a W-I white-light source (100 mW/cm$^2$) under terrestrial conditions in the laboratory in a quartz glass cell. Samples for the tests in the Drop Tower facility were prepared one week prior to testing and stored under N$_2$ atmosphere in the dark. XPS analysis of the stored samples did not show changes of the surface composition in comparison to freshly prepared samples (see below).

**Fabrication of rhodium nanostructures**. SNL[16] was employed to fabricate rhodium nanostructures on the InP substrate. For creating the masks, mono-dispersed beads of polystyrene (PS) sized 784 nm obtained at a concentration of 5% (w/v) from Microparticles GmbH were dissolved in MiliQ water and further diluted. For the final solution of 600 µl, 300 µl of the PS-beads dispersion was mixed with 300 µl ethanol containing 1% (w/v) styrene and 0.1% sulphuric acid (v/v). The solution was applied onto the air–water interface using a Pasteur pipette with a curved tip. In order to raise the area of the monocrystalline structures, the petri dish was gently turned, resulting in the transformation of multiple smaller domains into larger ones. The solution was carefully distributed to cover 50% of the water surface with a hcp monolayer, while leaving place for stress relaxation and avoiding formation of cracks in the lattice during the next preparation steps. The photoelectrochemically conditioned p-InP electrodes were delicately placed under the floating closed-packed PS sphere mask in the petri dish. Residual water was gently removed by pumping and evaporation with the mask being subsequently deposited onto the electrode. After the surface was dried with N$_2$, rhodium was photoelectrochemically deposited through the PS spheres as described above. The samples were furthermore rinsed with MiliQ water and dried under a gentle flow of N$_2$. The PS spheres were removed from the surface by placing the electrodes for 20 min under gentle stirring in a beaker with toluene. The electrodes were further cleaned by rinsing the sample with acetone and ethanol for 20 s. In order to remove residual carbon from the surface, O$_2$-plasma cleaning was employed for 6 min at a process pressure of 0.16 mbar, 65 W and gas inflows of O$_2$ and Ar of 2 sccm and 1 sccm, respectively.

**Structural and optical characterization**. Soft Tapping Mode Atomic Force Microscopy (TM-AFM) was used for the characterization of the surface morphology after each treatment step using a Bruker Dimension Icon AFM. In order to optimize the tapping (mode) frequency and experimental parameters such as gain, set point and cantilever tuning, ScanAsyst mode was used. ScanAsyst-Air tips (silicon nitride) were employed with a rotated (symmetric) geometry and a nominal tip radius of 2 nm. Peakforce Quantitative Nanomechanical parameters provide information on the height, adhesion and deformation of the sample surface.

Reflectance spectra of the thin-film and nanostructured photoelectrodes were obtained in air using a Cary 5000 UV/vis/NIR with an integrating sphere that include diffuse reflectivity measurement.

SEM images were obtained with a FEI Nova NanoSEM 450 microscope.

HRTEM analysis was performed with a Philips CM-12 electron microscope with twin objective lenses as well as a CCD camera (Gatan) system and an Energy-dispersive spectroscopy of X-rays system to measure the sample composition. For sample preparation, the thin rhodium film deposited on the p-InP substrate was scratched off and placed onto an amorphous carbon-coated (ca. 50 Å thickness) copper grid. The grid was then transferred to an electron microscope. A point number of grids was prepared from each sample in order to ensure the reproducibility of the preparative procedure.

**Photoelectron spectroscopy**. XPS analysis was performed using a system from VG Scienta with a base pressure below $8 \times 10^{-9}$ mbar equipped with a Scienta R3000 analyser and a monochromatic Al $K_\alpha$ (1486.6 eV) X-ray source. The analyser was operating at a pass energy fixed at 200 eV for survey scans and 50 eV for regional spectra acquisition. The used slit sizes were 3 mm and 0.4 mm for the survey and region scans, respectively. The measured surface area was $5 \times 1$ mm$^2$.

The binding energy scale was calibrated by calibrating the position of the C 1$s$ peak at 284.8 eV. The background photoelectron intensity was subtracted by the Shirley method[27]. The area under the principal peaks of each element in the XPS spectra and atomic sensitivity factors were used for calculations of atomic concentrations of the elements in approximately top 3–12 nm of the sample surface, depending on a sample.

Prior to testing the p-InP–Rh photocathodes in the drop tower, it was investigated whether they could be prepared in advance of the drop experiments and then stored under nitrogen atmosphere until used. Two samples were prepared and stored under nitrogen atmosphere by 4 days. The XPS spectra after four days of storage did not significantly change from the ones of freshly prepared samples. Furthermore, structural investigations of the electrode surface prior and after the drop did not suggest any changes of the surface morphology caused by the capsule deceleration process.

**Photoelectrochemical experiments in microgravity**. Microgravity environments were realized at the Drop Tower facility at the Centre of Applied Space Technology and Microgravity (ZARM), Bremen. With the ZARM Catapult System, 9.3 s of microgravity could be generated[18]. Here, the capsule was launched upwards from the bottom of the tower by a hydraulically controlled pneumatic piston-cylinder system and was decelerated again in a container which was placed onto the cylinder system during free fall of the capsule. The approached minimum g level was about $10^{-6}g$.

For the photoelectrochemical experiments in the drop tower, a custom-made two-compartment photoelectrochemical cell was used (filling volume of each cell: 250 mL). Each cell consisted of two optical windows made of quartz glass (diameter: 16 mm) through which the working electrode was illuminated. Photoelectrochemical measurements in the two cells were carried out in a three-electrode arrangement with a Pt counter electrode and an Ag/AgCl (3 M) reference electrode in HClO$_4$ (1 M) with the addition of 1% (v/v) isopropanol to reduce the surface tension of the electrolyte and favour gas bubble release. XPS measurements and photoelectrochemical measurements in terrestrial conditions did not show any effect of the isopropanol on the (photoelectrochemical) properties of the photoelectrodes. The light intensity of 70 mW/cm$^2$ was provided by a W-I white-light source (Edmund Optics). All experiments were carried out under ambient pressure.

Two cameras (Basler AG; acA2040-25gc and acA1300-60gm NIR, lens types: 35 mm Kowa LM35HC 1" Sensor F1.4 C-mount and Telecentric High Resolution Type WD110 series Type MML1-HR110, respectively) were attached to each cell via optical mirrors (monochromatic camera, side) and beamsplitters (colour camera, front, see Fig. 1c) to record the gas bubble formation in microgravity conditions. Data were stored during each drop on a Matrox 4Sight GPm integrated unit in the drop capsule. Single pictures were recorded at a frame rate of 25 fps (front camera) and 60 fps (side camera).

The photoelectrochemical set-up and the cameras were mounted on an optical board (Thorlabs) attached to the capsule. Power supply in the capsule was provided by a battery. Prior to each drop and during the evacuation time of the drop shaft (about 1.5 h), the capsule and the photoelectrochemical cell were set under Ar atmosphere which was maintained during the drop and during capsule recovery after the drop (about 45 min).

For the photoelectrochemical measurements, an automated drop sequence was written which was started prior to each drop. Upon reaching μg conditions, the sequence started cameras, illumination sources and potentiostats while simultaneously immersing the working electrode into the electrolyte using a pneumatic system (see Fig. 1 and Supplementary Figure 1 for more detailed information). Photoelectrochemical measurements such as cyclic voltammetry and chronoamperometric measurements were performed during the 9.3 s of microgravity. At the end of the drop, when the drop capsule was decelerated again to zero velocity, the sample was emersed from the electrolyte and the cameras, potentiostats and illumination source were switched off. The pneumatic system used for immersing and emersing the sample into and out of the electrolyte ensured that surface morphology changes of the electrode resulted not from long-term exposure to the electrolyte prior or after each drop. After retrieving the capsule from the deceleration container and removal of the protection shield, the samples were removed from the pneumatic stative, rinsed with MiliQ water and dried under nitrogen flux. The sample was stored under N$_2$ atmosphere until the optical and spectroscopic investigations were carried out.

**Theoretical simulations**. Lumerical FDTD, a commercial electromagnetic simulation software package, was used to optically model the system. To apply the above set of equations to the structures used here, the following set of assumptions is made. The current–voltage curve incorporating mass transport considerations (Eqs. (2) and (3)) is used for the thin-film sample in microgravity environments, and a limiting mass transport current density, $j_{mtl}$, of ±5 mA/cm$^2$ is assumed for

the anodic and cathodic current density respectively; for all other current–voltage curves Eq. (1) is used. These assumptions are based on experimental observations, as discussed above. The catalyst exchange current density, $j_{0,cat}$, is assumed to be 0.1 mA/cm$^2$, which is consistent with experimental reports in literature for Rh as a hydrogen evolution catalyst[8]. For the InP|Rh Schottky junction, the dark current ($j_0$) is assumed to be $10^{-8}$ mA/cm$^2$. Due to the InP$_x$O$_y$ layer, the ideal equations for the dark current of a Schottky junction did not accurately describe the system, therefore, this value is based on a fit to the experimentally measured current–voltage curves. For the thin-film sample under simulated microgravity conditions, $j_0$ was assumed to be $10^{-5}$ mA/cm$^2$, accounting for enhanced charge recombination processes in the semiconductor due to mass transfer limitations. The $f_{SA}$ values for the thin-film and nanostructured samples are 1.16 and 1.1, respectively, and are based on the surface areas of the catalyst as determined from the AFM data (Fig. 3b).

Due to the nanostructuring of our catalyst, numerical simulations are required to accurately determine the limiting photocurrent density, $j_L$, which is needed to apply the above set of equations to our photocatalytic system. Lumerical FDTD was used to obtain the InP absorption spectrum, $f_A(\lambda)$. The InP absorption spectrum was furthermore weighted with the lamp spectrum which was used in our experiment and via integration, the absorbed photocurrent, $j_L$, according to Eq. (4) was obtained. Here, $\lambda$ is the wavelength and $\lambda_{Eg}$ is the wavelength corresponding to the semiconductor band edge which is 925 nm for InP.

$$j_L = \int_0^{\lambda_{Eg}} f_A(\lambda) \cdot AM1.5G(\lambda) d\lambda \qquad (4)$$

In the optical simulations, the device structure is defined as a semi-infinite layer of InP coated with an 8 nm layer of InP$_x$O$_y$ and an effective medium layer of Rh (see XPS data discussion above and refs. [20,28]), all embedded in water. The Rh|H$_2$O effective medium layer is assumed to follow the Maxwell Garnett approximation, whereas the fill fraction of Rh was 0.4. For the thin-film and nanostructured samples, a layer thickness of 20 and 25 nm, respectively, was used. For the nanostructured Rh layer, the pattern is based on the assumption that the polystyrene spheres were hexagonally close-packed on the electrode surface with each sphere resulting in a cylindrical opening in the Rh layer, possessing a radius of 200 nm. These assumptions are based on previous publications (see above) and AFM data on the surfaces (Fig. 3b).

**Data availability**. All relevant data are available from the authors upon request.

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

## Acknowledgements

K.B. acknowledges funding from the fellowship programme of the German National Academy of Sciences *Leopoldina*, grant LPDS 2016-06 and the European Space Agency. Furthermore, she would like to thank Dr. Leopold Summerer, the Advanced Concepts Team, Alan Dowson, Dr. Jack van Loon, Dr. Gabor Milassin, Marcel van Slogteren and Dr. Robert Lindner (ESTEC), Robbert-Jan Noordam (Notese) and Prof. Harry B. Gray (Caltech) for their great support. M.H.R. is grateful for generous support from Prof. Nathan S. Lewis (Caltech). K.B. and M.H.R. acknowledge support from the Beckman Institute of the California Institute of Technology and the Molecular Materials Research Center. M.G. acknowledges funding from the Guangdong Innovative and Entrepreneurial Team Program titled 'Plasmonic Nanomaterials and Quantum Dots for Light Management in Optoelectronic Devices' (No. 2016ZT06C517). Furthermore, the author team greatly acknowledges the effort and support from the ZARM Team with Dr. Thorben Könemann and Dr. Martin Castillo at the Bremen Drop Tower. It is also thankful for enlightening discussions with Prof. Yasuhiro Fukunaka (Waseda University), Prof. Hisayoshi Matsushima (Hokkaido University) and Dr. Slobodan Mitrovic (Lam Research). The team would also like to thank Dr. Eser Metin Akinoglu from the International Academy of Optoelectronics, Zhaoqing, for his help with the SEM characterization of the samples and Dr. Axel Knop-Gericke (Fritz Haber Institute of the Max Planck Society) for his generous help with XPS measurements.

## Author contributions

K.B., M.H.R., J.L. and H.-J.L. planned and carried out the terrestrial experiments and the experiments at the Bremen Drop Tower. Ö.A., K.B. and J.L. prepared the nanostructured photoelectrodes under the supervision of M.G. and H.-J.L. K.T.F. carried out the theoretical calculations and simulations. K.B., H.-J.L and K.T.F. wrote the manuscript which is approved by all authors.

## Additional information

**Competing interests:** The authors declare no competing interests.

