## [Peer Review File · Nature Communications]

Reviewers' comments:

Reviewer #1 (Remarks to the Author):

This paper reported efficient light-generated production of hydrogen on InP-Rh photoelectrodes in microgravity environment by nanostructured catalyst. And the behavior of the applied system in terrestrial and microgravity environment is simulated using a kinetic transport model. The microgravity experiment adds valuable information to the solar hydrogen generation research field. However, it would be better if the questions as below could be addressed before publication.

1. The whole theory is based on the assumption that the formation of a forth layer by the evolved hydrogen bubbles on the electrode surface that hinders mass transfer. However, how to quantify the forth layer remains unclear. Simply judging by eye, under microgravity condition there are also a lot of bubbles attach to the surface of the nanostructured photoelectrode. The photocurrent of the thin film sample was reduced by 4 times, but it seems only a small portion of the electrode surface was covered by bubbles under microgravity. Is there a better way to characterize the properties (thickness? coverage?) of the forth layer?
2. Why is the open-circle voltage of the nanostructured photoelectrodes greatly ($\sim 100\text{mV}$) improved under microgravity environment, as shown in Fig. 2a?
3. Comparing the fabrication processes of thin film and nanostructured photoelectrode, it seems that the nanostructured sample has an additional O₂-plasma treatment process. This probably explains why the InO_x/PO_x signals are stronger in the nanostructured sample. I am wondering if the O₂-plasma treatment would change the surface chemistry of the photoelectrode that affects its microgravity behavior? To confirm/exclude this effect, O₂-plasma treated thin film photoelectrode should also be tested.
4. AFM images in Figure 3c should be given in the same scale for easy comparison.
5. Page 10, the open circuit voltage in SI Fig. 5b is only reduced by 50 mV, not 0.5 V.?
6. Electrochemical impedance spectroscopy (EIS) of the photoelectrodes in microgravity environment could be provided useful information on mass transportation. However, is that technically challenging given that the free fall time is only 9.3 s?
7. Chronoamperometry that capture the whole free fall process (before-during-after) should give some interesting dynamic results.
8. Equation 1, j_L was not defined.
9. Page 13, NSL should be SNL.
10. The addition of IPA to the HClO₄ electrolyte was not mentioned in the experimental section. What is the purpose of adding IPA? Would that change the PEC properties of the photoelectrodes?
11. Since pressure also plays a significant role in the bubble evolution, is the PEC cell under ambient pressure during the experiment?

Reviewer #2 (Remarks to the Author):

This work is appropriate to be published in the Nature Communications Journal but needs few re-arrangements.

The bibliography is rather complete concerning PhotoElectrochemistry but sufficient concerning general electrochemistry for hydrogen production under zero gravity.

In my opinion, 2 references might be added:

Y. Fukunaka, T. Homma, R. Hagiwara, T. Nohira, K. Hachiya, T. Matsuoka, Yunfeng Liang, T. Goto, H. Yasuda, H. Matsushima, N. Kishimoto, T. Ito, Y. Takahashi, K. Kinoshita, M. Takayanagi, Y. Sone, T. Ishikawa, T. Wakatsuki, K. Nishikawa, S. Yoda, M. Rosso, R. C. Alkire, W. Schwarzacher, O.

Magnussen, S. Kjelstrup, Ph. Mandin, D. R. Sadoway, R. C. Miranda, Non-equilibrium Electrochemical Processing of Nano-structured Energy Conversion & Storage Devices, Space Utiliz. Res., 27, 227-230 (2011)

Z. Derhoumi, Ph. Mandin, H. Roustan and, R. Wuthrich, Experimental investigation of two-phase electrolysis processes: comparison with or without gravity, J Appl Electrochem (2013) 43:1145–1161

In a concrete way:

Page7: Figure 2a and b are not clear.

Figure2a is not readable: the legend and caption for the 4 different cases : 1 g, 10⁻⁶g, thin film, nanostructured is not clear at all.. It seems that different shade of greys might allow in your opinion a easy reading? In fact With figure 2a I deduce that there is no influence of zero gravity on performance.. but I must be wrong?

Concerning Fig2b, the 1g thin film photograph is not planar? Difficult to see anything with these figures.

Page8: The influence of Surface Nanotopography might be given first in my opinion...

Generally it is clearer to explain the experimental set up and the design of experiments. In this paper, the experimental set-up is given after...

Figure3b is really strange for me: the line "thin film" gives information at "500 nm scale" whereas the line "nanostructured" gives information at "2 and 1 μm ".. I might have think that it should be the contrary? Am I wrong?

Page9: you pass directly from fig3 to fig5 instead of the fig4 introduction? Why?

At the end of Page9 "Output characteristic simulation" it is introduced an other experimental set-up (in a numerical part?)... no scheme is given?

Page10-11: I don't know if it is my own pdf editor but all equations (1), (2), (3), (4) page20 ...were not correctly printed

Page12: Figure4 has the same problem as Figue2: no clear legend and caption..

Impossible to determine the 4 cases with only grey shades... In text you speak of yellow or cyan lines which are not allowed with usual grey scale printers.

I think these simulations should have been compared with experimental results of figure 2..But not in a clear way: difference theory/experiments...?

Page15: after the part "conclusion" arrives a big experimental set-up "materials and methods" part.. I think one part might be put at the beginning and perhaps the rest in a correctly named "Annexe"...?

Reviewer #3 (Remarks to the Author):

The manuscript "Efficient Solar Hydrogen Generation in Microgravity Environment" explores how microgravity conditions affect the photoelectrochemical performance of InP/Rh photocathodes during Hydrogen evolution reaction. Concretely, the authors observed that flat films were highly sensitive to the microgravity conditions experiencing a drastic decrease in photocurrent and photovoltage, whereas the performance of nanotextured films remained unaltered. As the authors highlighted in the introduction, several studies have addressed and examined the performance of electrolyzers under these so-called microgravity conditions, but none of them dealt with photoelectrodes for direct water splitting. These reports indicated that the lack of buoyancy in these conditions was preventing from achieving a performance close to that obtain in "standard conditions" given the limitations in mass transfer imposed in the surface of the electrode when the bubbles of evolving gas blocked the catalytic surface. Obviously, when moving from "dark electrocatalyst" (electrolyzer) to "photoelectrodes" the authors found that, again, the lack of buoyancy in microgravity was the main responsible for the decreased performance, although the authors offer an interesting approach to address this known issue, nanotexturing the catalytic surface to promote the desorption of the gas bubbles, and beside they develop a simple model to simulate the results. The manuscript is well-written and structured, and provides an interesting route to address the formation of "froth layers" on the catalytic surface under microgravity. The novelty and impact of this work certainly deserves further consideration for publication in Nature Communications after a minor revision:

- In Figure 2b the authors included some pictures to demonstrate that the formation of a "froth layer" is the main cause for the detrimental performance in the "thin film" with respect to the "nanostructured". However, if we have to consider as a direct proof the pictures corresponding to 10- 6g, it is not really evident that the "froth layer" of the thin film is more dense or blocking more surface area than the one depicted for the nanostructured one. Could the authors provide another picture where the differences are more evident?
- Regarding Figure 2a, the authors describe how the J-V curve decreased significantly for the thin film but, they should include some comments on why the Voc of the nanostructured one is further increase under microgravity conditions (it shifts about > 100 mV under microgravity).
- Regarding the electrocatalyst, the authors indicate in page 8 that in both "flat" and "nanostructured" Rh electrocatalyst deposition the coverage is similar "the Rh 3d3/2 and 3d5/7 signal intensities are almost identical, suggesting a similar coverage of Rh on both electrodes". This sentence could be misleading since clearly the coverage of the underlying InP is different according to the AFM and SEM images. I guess the authors refer to the same catalyst loading?

Expectedly this could also be correlated to the integrated current during the electrochemical deposition of Rh.

- The authors established simple models that, quite successfully, describe the performance in microgravity. However, it would certainly add more impact to this work if the authors modify their model/equations to incorporate a parameter that takes into account/evaluate the loss of voltage caused by the froth layer.

- Regarding the better bubble desorption in the nanostructure the authors could include the reference *Angew. Chem. Int. Ed.* 2012, 51, 10760 where the use of nanopillars of InP demonstrated a fast desorption of H₂ bubbles, or others.

Response to reviewer comments

Reviewer #1

1. *The photocurrent of the thin film sample was reduced by 4 times, but it seems only a small portion of the electrode surface was covered by bubbles under microgravity. Is there a better way to characterize the properties (thickness? coverage?) of the froth layer?*

This is a valuable comment made by the reviewer; we include a different image of the thin film photoelectrode in microgravity environment in the revised manuscript, which shows the described phenomena in a clearer way. It is, however, difficult to completely describe the thickness of the froth layer and its coverage of the electrode in the recorded images; recently, Zhang et al. (2006) have demonstrated the formation of so-called “nanobubbles” on the electrode surface, which are also formed during electrochemical gas bubble formation in terrestrial environments (Zhang et al. *Langmuir* **22**, 8109-8113, 2006). These “nanobubbles” can e.g., be detected by in-situ electrochemical AFM measurements where bubble nucleation and growth can be investigated on a nm scale and the total surface coverage of gas bubbles on the electrode can be estimated. These measurements would certainly be of great advantage here and terrestrial experiments with thin film and nanostructured photoelectrodes are planned. In microgravity environment, we are unfortunately limited to the camera resolution and can therefore only determine the gas bubble froth layer thickness on the μm to mm scale.

2. *Why is the open-circuit voltage of the nanostructured photoelectrodes greatly (~100mV) improved under microgravity environment, as shown in Fig. 2a?*

We appreciate the comment and include a study based on five different thin film and nanostructured photoelectrodes in the SI, where we measured in independent cyclic voltammetry experiments the V_{OC} of the samples. As Table 1 (SI) shows, the V_{OC} of the nanostructured and the thin film electrode is almost identical under terrestrial conditions and the V_{OC} value of the nanostructured photoelectrode measured under terrestrial conditions in Figure 2a is in the error range.

3. *I am wondering if the O_2 -plasma treatment would change the surface chemistry of the photoelectrode that affects its microgravity behavior? To confirm/exclude this effect, O_2 -plasma treated thin film photoelectrode should also be tested.*

This is an interesting question, however, the O_2 plasma treatment is likely to improve the stability of the open InP spots on the nanostructured photoelectrode which is exposed directly to the electrolyte in the experiments. The formed InO_x is likely to protect the surface, although the catalytic event occurs at the Rh catalyst and the directly underlying InP. Therefore, the photoelectrocatalytic event is very unlikely to be affected by the O_2 plasma treatment.

4. *AFM images in Figure 3c should be given in the same scale for easy comparison.*

The reason for the two resolutions of the AFM images - 500nm for the thin film photoelectrode and $1\mu\text{m}$ for the nanostructured sample - is the better visibility of the

fine structure of the deposited Rh of the thin film sample at this resolution and the periodic arrangement of the nanostructured photoelectrode at $1\mu\text{m}$. Nevertheless, we included an AFM image with a 500nm resolution of the nanostructured sample in the SI.

5. *Page 10, the open circuit voltage in SI Fig. 5b is only reduced by 50 mV, not 0.5 V.?*

We appreciate the comment, this is correct. We changed the value accordingly in the revised manuscript.

6. *Electrochemical impedance spectroscopy (EIS) of the photoelectrodes in microgravity environment could be provided useful information on mass transportation. However, is that technically challenging given that the free fall time is only 9.3 s?*

EIS could be an interesting experiment, provided that more experimental time is available as rightfully addressed by the reviewer. This would be an experiment to be planned for parabolic flights or stationary microgravity (ISS).

7. *Chronoamperometry that captures the whole free fall process (before-during-after) should give some interesting dynamic results.*

This is an interesting comment of the reviewer; indeed, we recorded chronoamperometric data during the free fall experiment, which we now mention in the manuscript. We also include the corresponding data in the SI.

8. *Equation 1, j_L was not defined.*

Yes, this is correct, we only defined j_L in the *Materials and Methods* part. In the revised manuscript, we included a definition with equation 1.

9. *Page 13, NSL should be SNL.*

This is correct. We changed it in the revised manuscript.

10. *The addition of IPA to the HClO_4 electrolyte was not mentioned in the experimental section. What is the purpose of adding IPA? Would that change the PEC properties of the photoelectrodes?*

We greatly appreciate the comment; this is correct, we only included information on the addition of IPA in the figure caption of Figure 2 and in the text. In the revised manuscript, this information is also added in the experimental section and an explanation is included: IPA reduces the surface tension of the electrolyte, which favors gas bubble desorption according to Vogt H. *Electrochim. Acta* **56** 2404-2410 (2011). XPS measurements confirm that the surface composition of the photoelectrode was not affected by this addition to the electrolyte and also terrestrially, the photoelectrochemical properties were not influenced. In microgravity environment, however, we observed in a parallel study that the addition of 1% IPA leads to an increased gas bubble release and therefore, an improved J-V behaviour for both, thin film and nanostructured electrode.

11. *Since pressure also plays a significant role in the bubble evolution, is the PEC cell under ambient pressure during the experiment?*

Yes, the PEC cell is under ambient pressure during the experiment in microgravity conditions which is also mentioned in the revised manuscript (p. 19).

Reviewer #2

1. *In my opinion, 2 references might be added.*

We appreciate the comment; the two suggested references are added to the manuscript.

2. *Page 7: Figure 2a and b are not clear.*

Figure 2a is not readable: the legend and caption is not clear. Concerning Figure 2b, the 1g thin film photograph is not planar?

For a better visibility, independently of the employed color code, we introduced a dotted line for the JV behaviour of the thin film sample and a dashed line for the nanostructured sample in terrestrial conditions and a dashed-dotted line for the thin film sample and a straight line for nanostructured electrode in microgravity environment.

Yes, it is correct, the so-called “thin film” electrode is not strictly planar and the photoelectrochemically deposited Rh shows an AFM height profile with variations in the range of 13nm, which is also mentioned in the manuscript on p. 15. In comparison to the “nanostructured” photoelectrode, however, the “thin film” electrode exhibits a continuous thin film of deposited rhodium.

3. *Page 8: The influence of Surface Nanotopography might be given first in my opinion.*

We appreciate the suggestion made by the reviewer, we followed, however, the order of the experimental investigations, which started with the experiments at the drop tower and the observation of the differences in the JV behaviour of the thin film and nanostructured photoelectrodes in microgravity environment. Subsequently, we investigated and characterized the electrodes afterwards to further understand the role of the electrocatalyst surface topography for the photoelectrochemical experiments in microgravity.

4. *Generally, it is clearer to explain the experimental set up and the design of experiments. In this paper, the experimental set-up is given after.*

This is a reasonable suggestion, we followed, however, the *Nat. Comm.* manuscript guidelines, which also accounts for comment 11. Nevertheless, the first part of the results section, includes a detailed description of the experiment in microgravity environment. Additional information related to the experimental part can also be found in Figure 1.

5. *Figure 3b, the line “thin film” gives information at “500 nm scale” whereas the line*

“nanostructured” gives information at “2 and 1 μm ”. I might have think that it should be the contrary?

We are grateful for the comment and would like to refer the reviewer to our answer to the comment #4 of reviewer #1.

6. *Page 9: you pass directly from Figure 3 to Figure 5 instead of the Figure 4 introduction?*

Since we included more SI Figures in the revised manuscript, the numeration of the figures changed slightly. Nevertheless, we appreciate the comment, although we kept the original structure of the manuscript. After the introduction of Figure 3, showing the surface characteristics of the two photoelectrodes, these characteristics are compared to the surface compositions determined by XPS, the spectra are shown in Figure 5 in the SI. Thereafter, in the discussion part, the drop tower results and the theoretical assumption of mass transfer limitations influencing the JV characteristics of the thin film sample are compared to an experimental series under terrestrial conditions, where the influence of the mass transfer limitation on the JV characteristics is demonstrated. The results are presented in the SI Figure 6. In the following, Figure 4 in the main paper is introduced, where the observed JV behaviors under terrestrial and microgravity conditions is simulated.

7. *At the end of Page 9 “Output characteristic simulation” it is introduced another experimental set-up (in a numerical part?). No scheme is given?*

We appreciate the comment, however, since the theoretical simulations are based on previous publications which illustrate the applied method and procedures well, we would like to refer the reviewer to the cited references for further details of the theoretical modelling.

8. *Page 10-11: I don't know if it is my own pdf editor but all equations (1), (2), (3), (4) page 20 were not correctly printed.*

We apologize for any inconveniences caused by the display of the equations. Hopefully, the pdf version of the revised manuscript allows the complete display of the equations.

9. *Page 12: Figure 4 has the same problem as Figure 2: no clear legend and caption. Impossible to determine the 4 cases with only grey shades. In the text, you speak of yellow or cyan lines which are not allowed with usual grey scale printers.*

We would like to refer the reviewer to our reply to comment #2; we introduced additional graphical features which should allow the distinction of the different JV curves independently from the color code.

10. *I think these simulations should have been compared with experimental results of Figure 2. But not in a clear way: difference theory/experiments...?*

We appreciate the comment and included an additional discussion part in the revised manuscript, referring to differences in the theoretical simulation and the experiment.

11. *Page 15: after the part “conclusion” arrives a big experimental set-up “materials and methods” part. I think one part might be put at the beginning and perhaps the rest in a correctly named “Annex”?*

We would like to refer the reviewer to our reply to comment #4.

Reviewer #3

1. *In Figure 2b the authors included some pictures to demonstrate that the formation of a “froth layer” is the main cause for the detrimental performance in the “thin film” with respect to the “nanostructured”. However, if we have to consider as a direct proof the pictures corresponding to $10^{-6}g$, it is not really evident that the “froth layer” of the thin film is denser or blocking more surface area than the one depicted for the nanostructured one. Could the authors provide another picture where the differences are more evident?*

We are grateful for the comment and would like to refer the reviewer to our reply to reviewer #1, comment #1.

2. *Regarding Figure 2a, the authors describe how the J-V curve decreased significantly for the thin film but, they should include some comments on why the V_{oc} of the nanostructured one is further increased under microgravity conditions.*

We greatly appreciate the comment and would also like to refer the reviewer again to our reply to reviewer #1, comment #2.

3. *Regarding the electrocatalyst, the authors indicate in page 8 that in both “flat” and “nanostructured” Rh electrocatalyst deposition the coverage is similar “the Rh $3d_{3/2}$ and $3d_{5/2}$ signal intensities are almost identical, suggesting a similar coverage of Rh on both electrodes”. This sentence could be misleading since clearly the coverage of the underlying InP is different according to the AFM and SEM images. I guess the authors refer to the same catalyst loading? Expectedly this could also be correlated to the integrated current during the electrochemical deposition of Rh.*

This is a valuable comment which is very much appreciated. It is true that the indicated sentence is misleading; the signal intensities are similar, although the local Rh coverage is distinctively different due to the different surface topographies of the Rh catalyst. We changed the sentence in the revised manuscript to: “Despite the fact that Rh was deposited through the polystyrene sphere mask, the Rh $3d_{3/2}$ and $3d_{5/2}$ signal intensities are almost identical, suggesting a similar overall coverage of Rh on both electrodes with distinctive differences in the local coverage due to the different surface topographies.” (p. 9)

4. *The authors established simple models that, quite successfully, describe the*

performance in microgravity. However, it would certainly add more impact to this work if the authors modify their model/equations to incorporate a parameter that takes into account/evaluate the loss of voltage caused by the froth layer.

We are grateful for the comment; the voltage loss due to the froth layer is already taken into account by the extension of the Butler-Volmer equation for mass transfer limitation. For the description of the voltage loss due to recombination processes, however, the so-called “saturation current” or “dark current” plays a crucial role for photodiodes. We succeeded in modelling the JV behaviour of the thin film electrode in microgravity environment considering mass transfer limitations and resulting increased surface recombination of charge carriers by lowering the saturation current. Please compare Figure 4 in the revised manuscript and the *Materials and Methods* part for further details.

5. *Regarding the better bubble desorption in the nanostructure the authors could include the reference Angew. Chem. Int. Ed. 2012, 51, 10760 where the use of nanopillars of InP demonstrated a fast desorption of H₂ bubbles, or others.*

We appreciate the suggestion and include the additional reference in the revised manuscript.

REVIEWERS' COMMENTS:

Reviewer #1 (Remarks to the Author):

I feel that the points raised in the previous round of review have been satisfactorily addressed.

Reviewer #2 (Remarks to the Author):

Thank you for all the answers and correction made. On my side all is better.
regards

Reviewer #3 (Remarks to the Author):

The authors have addressed compellingly all the comments, and I would suggest to accept the manuscript for publication.